# Harnessing the Nucleolar DNA Damage Response in Cancer Therapy

**DOI:** 10.3390/genes12081156

**Published:** 2021-07-28

**Authors:** Jiachen Xuan, Kezia Gitareja, Natalie Brajanovski, Elaine Sanij

**Affiliations:** 1Peter MacCallum Cancer Centre, Melbourne, VIC 3000, Australia; anthony.xuan@petermac.org (J.X.); Kezia.Gitareja@petermac.org (K.G.); Natalie.Brajanovski@petermac.org (N.B.); 2Sir Peter MacCallum Department of Oncology, University of Melbourne, Melbourne, VIC 3010, Australia; 3Department of Clinical Pathology, University of Melbourne, Melbourne, VIC 3010, Australia; 4St Vincent’s Institute of Medical Research, Fitzroy, VIC 3065, Australia; 5Department of Medicine -St Vincent’s Hospital, University of Melbourne, Melbourne, VIC 3010, Australia

**Keywords:** rDNA, RNA polymerase I, nucleolus, DNA damage response, CX-5461

## Abstract

The nucleoli are subdomains of the nucleus that form around actively transcribed ribosomal RNA (rRNA) genes. They serve as the site of rRNA synthesis and processing, and ribosome assembly. There are 400–600 copies of rRNA genes (rDNA) in human cells and their highly repetitive and transcribed nature poses a challenge for DNA repair and replication machineries. It is only in the last 7 years that the DNA damage response and processes of DNA repair at the rDNA repeats have been recognized to be unique and distinct from the classic response to DNA damage in the nucleoplasm. In the last decade, the nucleolus has also emerged as a central hub for coordinating responses to stress via sequestering tumor suppressors, DNA repair and cell cycle factors until they are required for their functional role in the nucleoplasm. In this review, we focus on features of the rDNA repeats that make them highly vulnerable to DNA damage and the mechanisms by which rDNA damage is repaired. We highlight the molecular consequences of rDNA damage including activation of the nucleolar DNA damage response, which is emerging as a unique response that can be exploited in anti-cancer therapy. In this review, we focus on CX-5461, a novel inhibitor of Pol I transcription that induces the nucleolar DNA damage response and is showing increasing promise in clinical investigations.

## 1. Introduction

RNA polymerase I (Pol I) transcribes the rRNA (rRNA) genes (rDNA) to produce the 47S precursor-rRNA (pre-rRNA) within the nucleoli. The pre-rRNA is subsequently processed by multiple endonucleases to form the mature 18S, 5.8S, and 28S rRNAs [1,2,3,4]. Together with the 5S rRNA, these rRNAs form the nucleic acid backbone of the ribosomes, and as such Pol I transcription dictates the rate of ribosome biogenesis and mRNA translational capacity [5].

Pol I transcription underpins the structure of the nucleoli, the site of Pol I transcription and ribosome biogenesis. However, the nucleoli play important additional functions, notably as a central hub for coordinating cellular response to stress. Indeed, the integrity of the nucleolus can modulate cellular homeostasis beyond ribosome biogenesis [6]. Such roles include titration of tumour suppressors and oncogenes, modulating the DNA damage response (DDR) and DNA repair, modulating cell cycle response, stress responses, global gene expression and maintaining genomic stability [6].

In this review, we discuss rDNA features that render the rDNA loci as vulnerable genomic regions and hotspots for DNA damage and recombination events in cancer. We focus on rDNA damage repair and explore the rationale for targeting rDNA instability and Pol I transcription as a potential treatment strategy to combat cancer. We discuss the consequences of inhibiting Pol I transcription and the crosstalk between the nucleolus and the DDR. In fact, many classical chemotherapeutic agents (e.g., oxaliplatin, cisplatin, actinomycin D) have been discovered to act through distinct mechanisms of action that include inhibition of Pol I transcription and ribosome biogenesis [7,8,9,10]. The development of a number of less genotoxic drugs that selectively target Pol I transcription has established a new paradigm for cancer therapy [11]. In particular, we expand on the therapeutic potential of the first-in-class inhibitor of Pol I transcription CX-5461, through its ability to activate the nucleolar DNA damage response (n-DDR). The analysis of CX-5461′s mode of action and therapeutic efficacy gives invaluable insights into the design of optimal combination therapy approaches that enhance the efficacy of current standard-of-care cancer therapies.

## 2. rDNA Structure

The nucleolus is a non-membranous organelle, that forms at the end of mitosis around nucleolar organizer regions (NORs), which consist of tandemly repeated arrays of rDNA repeats (Figure 1A). The cytoarchitectural upstream binding factor (UBF) binds to rDNA and is responsible for bookmarking NORs for post-mitotic nucleolar formation by decondensing rDNA chromatin and establishing a euchromatin transcriptionally active rDNA chromatin state [12]. During interphase, two or more NORs coalesce to form multiple nucleoli in exponentially growing cells [13]. The number of rDNA units per cell varies greatly among eukaryotes, from 40 to ~19,000 in animals and correlates positively with genome size [5,14]. In human cells, there are up to 300 copies of rRNA genes per haploid genome, arranged in a head to tail orientation in clusters of tandem repeats. These rDNA clusters are dispersed on the p-arms of acrocentric chromosomes (13, 14, 15, 21 and 22) and contain ~70 copies of the rRNA genes per cluster [15]. A single rDNA unit, containing a transcribed region followed by intergenic spacer (IGS) is referred to as a canonical rDNA unit. The rDNA clusters are flanked by highly conserved proximal and distal junctions. The proximal junction located towards the centromeres is composed of segmentally duplicated sequences resembling centromeric regions on other chromosomes. The distal junction located towards the telomeres contains 48 bp satellite and large inverted repeats unique to the five acrocentric chromosomes. This distinctive sequence anchors to the nucleolar periphery, having importance in regulating the transcriptional status of linked rDNA repeats and nucleolar organization [16].

UBF-bound NORs are associated with rDNA hypomethylation, acetylated histones and transcriptionally permissive or active rDNA features. In contrast, NORs unbound to UBF are associated with heterochromatic histone modifications, CpG hypermethylation at promoter regions and transcriptionally inactive rDNA [17,18,19,20]. The active rDNA repeats are transcribed by Pol I to produce the 47S pre-rRNA, which is processed to form the mature 18S, 5.8S and 28S rRNAs. These rRNAs then assemble with 5S rRNA transcribed by Pol III and ribosomal proteins (RPs) to generate the 40S and 60S subunits in the nucleolus, before being exported into the cytoplasm to form functional ribosomes (Figure 1B).

The rDNA repeats are some of the most actively transcribed genes, accounting for 35–60% of all cellular transcription [21]. Due to their repetitive nature and high Pol I transcription rates, the rDNA loci are inherently unstable and have been shown to be increasingly susceptible to DNA damage and chromosomal recombination events resulting in large copy number variations [22,23]. Variation in rDNA copy numbers independent of Pol I transcription rate has been associated with cancer [24]. It has been proposed that reduction of rDNA silencing and/or rDNA instability underpins global genomic instability and tumorigenesis [25,26,27,28,29,30]. In this review, we discuss how eukaryotic cells respond to rDNA damage and rDNA instability and the emerging potential of targeting the nucleolar DDR and rDNA instability in cancer therapy.

### 2.1. rDNA: An Intrinsically Unstable Genomic Region 

Several unique characteristics of the rDNA repeats, including their repetitive nature, high Pol I transcription rates and high GC content, present risks of genomic instability at the rDNA loci [31]. The head-to-tail arrangement of rDNA copies on the five human acrocentric chromosomes allows intrachromosomal recombination between distant repeats on the same chromosome and interchromosomal recombination between repeats on different chromosomes at proximity within a nucleolus. Thus, dysregulations in rDNA copy number and genomic rearrangements are likely to occur [30,32]. Furthermore, high levels of transcription render the rDNA clusters vulnerable to transcription-replication conflicts. A head-on collision between the Pol transcription machinery and DNA replication forks triggers local accumulation of positive DNA supercoiling (Figure 2). This increases torsional stress and replication fork stalling within the rDNA region, thereby impinging upon both transcription and replication processes [33]. In addition, negative DNA supercoiling due to DNA unwinding during transcription promotes R-loop formation comprising of a highly stable nascent RNA molecule and template DNA strand hybrid and a displaced non-template DNA strand (single-stranded DNA (ssDNA)) [34]. G quadruplexes (G4s) structures can form in the displaced strand of an R loop. The formation of both G4s and R-loops is favored by regions with negative torsional tension such as active gene promoters [35]. The unmethylated CpG island in the rDNA promoters has increased propensity to form R-loops, which under physiological conditions, prevents DNA methylation of CpG islands by DNA methyltransferase DNMT3B1, thus facilitating rDNA transcription [36]. However, unscheduled R-loops and/or stabilization of G4 structures can impede Pol elongation, resulting in the pileup of Pol molecules, ultimately provoking replication fork slowing and stalling leading to DNA damage [34]. Protective mechanisms, such as the binding of the transcription termination factor (TTF-I) and replisome factor Timeless to replication fork barriers (RFBs) in the IGS regions (Figure 2) provide spatial separation between transcription and replication machineries and prevent potential clashes at early S phase to minimize rDNA instability [37]. Nevertheless, the rDNA clusters have high susceptibility to replication stress, DNA damage and genomic instability [24].

### 2.2. rDNA Instability in Cancer

rDNA instability and deregulation of rDNA copy numbers have been associated with cancer progression. Spontaneous alterations in rDNA organization were over 100-fold elevated in cells lacking Bloom Syndrome (BLM) protein, a RECQ helicase involved in homologous recombination (HR) DNA repair, and 10-fold elevated in cells lacking ATM (ataxia-telangiectasia, mutated) compared with wild-type controls [38]. These rDNA alteration phenotypes seem to correlate with the increased cancer predisposition reported in Bloom syndrome and ataxia-telangiectasia patients [38]. These results suggest that defects in DDR and DNA repair pathways can induce rDNA instability that may ultimately result in the development of cancer. Furthermore, studies have shown frequent rDNA rearrangements in lung and colorectal carcinomas and Hodgkin’s lymphoma [39,40,41]. Interestingly, both gain and loss of rDNA repeats have been linked to several cancers. Increases in rDNA copy numbers have been identified in gastric, colorectal and lung cancers [39,42,43]. It is plausible that the increase in rDNA copy number provides an advantage in cancer cells in supporting high rates of Pol I transcription and ribosome biogenesis. Alternatively, a high rDNA copy number may be associated with increased rates of collision between Pol I transcription and replication machineries leading to replication stress and genomic instability and cancer development [33]. In contrast, recent studies have identified lower rDNA copy numbers in several types of cancers compared with adjacent normal tissues [27,30]. Low rDNA copy numbers are associated with the inactivation of tumor suppressors PTEN and p53 [27,30]. Hematopoietic cancer stem cells with PTEN deletion and lower rDNA copy numbers exhibit high rates of rDNA transcription, protein synthesis and cell proliferation. These observations suggest the presence of compensatory mechanisms that maintain transcription output by increasing Pol I transcription rates of the remaining active rDNA repeats [38]. Additionally, decreased rDNA copy numbers provide cancers a competitive growth advantage by potentially reducing rDNA heterochromatin levels, which can alter the global epigenetic landscape, lead to genome-wide dysregulation of gene expression and faster replication of the genome [26,33]. However, both loss and gain of rDNA repeats have been reported in invasive ductal breast carcinoma [29]. Altogether, rDNA instability in the form of loss or gain in rDNA copy numbers, rDNA rearrangements and reductions of rDNA heterochromatin levels underpin global genomic instability, and that this can drive the etiology and progression of cancer [26,27,28,29,30].

## 3. Mechanisms of rDNA Repair

rDNA lesions in the nucleolus trigger a specialized nucleolar-DNA damage response (n-DDR) [44]. The initial step of the n-DDR is very similar to the classic DDR and involves the recognition of DNA damage by activation of ATM and Ataxia telangiectasia and Rad3-related (ATR) kinases. These kinases in turn phosphorylate a broad range of targets and initiate signaling cascades that regulate various cellular processes including DNA replication, transcription and cell cycle progression. Upon recognition of rDNA double-strand breaks (DSBs), the n-DDR induces ATM-mediated repression of Pol I transcription in the affected nucleolus and subsequent segregation of the damaged rDNA repeats and repair proteins into nucleolar caps at the periphery, which provide an optimal environment for rDNA repair [45,46,47]. Although, the factors involved in mediating inhibition of Pol I transcription upon rDNA damage remain unclear.

Previous findings identified the involvement of ATM, Nijmegen breakage syndrome 1 (NBS1) and mediator of DNA damage checkpoint protein 1 (MDC1) in Pol I silencing following the induction of DSBs at rDNA [48,49]. However, recent studies have shown that MDC1 depletion has negligible effects on rDNA transcriptional inhibition and nucleolar recruitment of NBS1 after rDNA damage [46]. The ATM-activated nucleolar protein Treacle (also known as TCOF1) has been shown to be involved in recruiting the MRN complex comprising of NSB1, MRE11 and RAD50 to rDNA DSBs in the nucleolus [46,50,51]. Treacle also recruits DNA topoisomerase II binding protein 1 (TOPBP1) to activate ATR, resulting in complete inhibition of Pol I transcription and formation of replication protein A (RPA) foci in the nucleolus, indicative of the presence of ssDNA [52]. Moreover, several proteins including checkpoint kinase 1 and 2 (CHK1/2), MST2 kinase, cohesin subunits SMC1 and SMC3 have also been found to compromise Pol I transcriptional activity upon rDNA damage [52,53,54]. Persistent transcriptional inhibition triggers nucleolar restructuring, cap formation, mobilization of DSBs and recruitment of n-DDR repair proteins to the nucleolar caps [44,47]. However, a recent study has shown that ion micro-irradiation that specifically induced localized rDNA damage to a subnucleolar region, locally restricted transcriptional inhibition but did not lead to nucleolar segregation [55]. These findings suggest that both the magnitude and/or the type of rDNA damage can lead to different structural outcomes with respect to nucleolar organization and the choice of DNA repair pathway initiation. 

As per other regions of the genome, rDNA DSBs are primarily repaired by the HR and non-homologous end-joining (NHEJ) repair pathways. Whether the HR or the NHEJ pathways are utilized is dependent on the extent of rDNA damage and persistence of DSBs. Indeed, the transits from low levels of rDNA damage and partial transcriptional inhibition to ATM-dependent rDNA silencing and large-scale reorganization of nucleolar architecture is associated with a shift from NHEJ to HR type of repair [44] (Figure 3).

NHEJ is an efficient, but highly mutagenic process that can occur at all stages of the cell cycle. It is initiated by the binding of a Ku70/80 heterodimer to each end of a DSB, which in turn promotes the assembly of the DNA-dependent protein kinase catalytic subunit (DNA-PK) and allows synaptic complex formation. DNA termini unable to be ligated are removed or filled-in wjkith complementary single-stranded sequence before DNA Ligase IV and the X-ray repair cross-complementing protein 4 (XRCC4) complex facilitates ligation of the break. In a study involving I-PpoI endonuclease-induced DSBs at the 28S sequences of rDNA, DNA-PK inhibition leads to prolonged rDNA transcriptional repression and inefficient repair of DSBs [45]. Depletion of other NHEJ proteins also increased the severity of damaged rDNA loci. These data suggest NHEJ as the predominant mode of immediate rDNA DSBs repair within the nucleolar interior, mitigating the impact on Pol I activity [45]. However, persistent rDNA DSBs result in ATM-mediated transcriptional inhibition of the entire nucleolus and cell cycle-independent nucleolar restructuring and cap formation [44,45]. Damaged rDNA translocate to nucleolar caps, where HR repair proteins are present in abundance [56]. Various factors, such as RPA, RAD51, RAD52, BRCA1 and BRCA2, are then recruited to the nucleolar caps to repair rDNA in S/G2 and even in the G1 phase of the cell cycle, which is a unique feature of HR repair of rDNA [47]. Furthermore, whilst segregation of damaged rDNA repeats at the nucleolar caps can serve as a mechanism of physical separation from other chromosomes to prevent inter-recombination with other rDNA loci [57], HR repair has been reported to cause a reduction in rDNA copy number and cellular viability [56]. Depletion of HR factors rescued rDNA repeat loss and increased viability, indicating that HR repair of the repetitive rDNA can have deleterious consequences on genome stability and cell viability [56]. Further investigations into the mechanisms of rDNA repair and the consequences of rDNA damage are important in order to understand the role of rDNA instability in cancer development.

## 4. The Crosstalk between the Nucleolus and the DDR

Comprehensive analysis of the nucleolar proteome has identified over 4500 proteins to reside in the nucleolus [58,59,60] with only 30% of them identified to be associated with ribosome biogenesis (RPs; RNA-binding proteins; RNA helicases; RNA-modifying and related protein) [6,61]. The nucleolus is a central hub for coordinating multiple biological processes including cell cycle regulation, DNA repair and replication and stress signaling via mediating the sequestration or release of various factors involved in these processes [59,60]. The protein content of the nucleolus has been shown to be dynamic, changing dramatically under various stress conditions [6]. It is now clear that the nucleolar proteome undergoes distinct spatial and temporal alterations in response to different stress insults, suggesting that the nucleolus responds to different stress stimuli in a unique and specific manner [62]. Analyses of the nucleolar proteome have revealed that at least 166 unique DNA repair proteins reside in the nucleoli. Some of these factors are important for rDNA repair while others are proposed to be sequestered in the nucleoli until they are required for their functional role in the nucleoplasm [63]. However, these two scenarios may not be mutually exclusive.

### 4.1. The Nucleolus Is a Hub for DNA Repair Factors

40% of Poly-ADP ribose polymerase-1 (PARP1) enzyme, which is involved in the repair of single-stranded DNA breaks (SDBs) and base excision repair (BER), can be found in the nucleolus [64]. PARP1 is activated by binding to DNA breaks and it contributes to repair pathway choice and the efficiency of repair through modulation of chromatin structure and ADP-ribose posttranslational modifications of a multitude of DNA repair factors [65]. PARP1 is known to participate in various DNA repair pathways and in mediating replication fork reversal upon stalling of replication forks. It has also been implicated in the regulation of the microhomology-mediated end-joining (MMEJ) pathway [66], which utilizes short homologous sequences internal to the break ends to align them for repair [67]. Given the repetitive nature of rDNA, it is possible that PARP1 can mediate MMEJ repair of rDNA in addition to NHEJ and HR. However, several studies have documented various roles for PARP1 in ribosome biogenesis. As such, its role in the nucleoli may not be restricted to rDNA repair. Indeed, PARP1 has been implicated in the maintenance of the silent heterochromatic rDNA fraction [68], known to be important for maintaining rDNA stability. PARP1 interacts with TIP5, a subunit of the nucleolar remodeling complex (NoRC) and mediates the recruitment of DNA methyltransferase and histone deacetylase to rDNA promoters [69]. PARP1′s interaction with TIP5 leads to PARP1 auto-modification and subsequent TIP5 and/or histone ADPRylation, repression of Pol I transcription and rDNA silencing [69]. More recently, snoRNA-activated PARP1 was shown to ADP-ribosylates DDX21, an RNA helicase that localizes to nucleoli and promotes rDNA transcription when ADP-ribosylated [70]. It is plausible that PARP1′s role at rDNA may be influenced by the chromatin states of the rDNA repeats (Figure 4).

PARP1 interacts with several RecQ-like helicases including Werner syndrome RecQ such as helicase (WRN) and BLM [71], which also reside in the nucleolus and are implicated in resolving G4 structures, R-loops and stalled replication forks [72]. WRN and BLM interact with Pol I and facilitate Pol I transcription, suggesting their dual role in mediating Pol I transcription as well as resolving replication and topological stress at the rDNA loci. Similarly, Apurinic/Apryimidinic endonuclease 1 (APEX1) is another BER pathway factor that localizes to the nucleolus and is involved in both DNA repair and ribosome biogenesis. It is the primary endonuclease that allows for repair after the identification and removal of a damaged base [73]. In addition to its DNA repair function, strong evidence also suggests that it localizes to rDNA and interacts with nucleophosmin (NPM1), a multifunctional protein with various roles in ribosome biogenesis [74]. APEX1 also interacts with the 47S pre-rRNA and other rRNA species, suggesting a role for APEX1 at rDNA in repairing rRNA for quality control purposes [75]. Moreover, ATM and ATR, which are recruited to the nucleoli in response to DSBs and replication stress at the rDNA repeats, have been shown to interact with an extensive network of proteins involved in protein trafficking, RNA processing and transcription [76]. Taken together, the data strongly supports dual functional roles for DNA repair proteins in both the repair of rDNA and in ribosome biogenesis (Figure 4).

### 4.2. DNA Damage-Induced Nucleolar-Nucleoplasmic Shuttling

Stress responses to persistent DNA damage at the rDNA repeats involves halting of transcriptional activity and restructuring of the nucleoli. Damaged rDNA localizes to the nucleolar caps where DDR factors are recruited to promote rDNA repair. How the nucleolar proteome is altered in response rDNA damage remains unknown. As mentioned above, Treacle mediates inhibition of Pol I transcription by recruiting NBS1 to the nucleolus in a manner that is partly dependent on PARP1 activity [46]. On the other hand, a recent study suggests a reduction in PARP1 levels at sites of rDNA damage [55]. This is in contrast to PARP1′s accumulation that is usually observed at nucleoplasmic DNA damage sites [55]. The release of PARP1 from the nucleolus may possibly be a coordinated response that links n-DDR with global DDR. Indeed, PARP1 was recently found to mediate WRN and XRCC1 translocation from the nucleolus to the nucleoplasm upon treatment with H_2_O_2_ and the alkylating agent 2-chloroethyl ethyl sulfide [71]. Further investigations are required to understand nucleolar-cytoplasmic shuttling of DDR factors upon nucleolar reorganization in response to rDNA damage and/ or global DDR.

In relation to this, a landmark study by Rubbi and Milner (2003) [77] proposed that disruption of nucleolar structure and function and subsequent release of nucleolar components into the nucleoplasm is a common feature in most p53-inducing stresses (Reviewed in [6]). The nucleolar surveillance pathway describes the molecular response to perturbations in ribosome biogenesis and nucleolar structure, that are sensed by the nucleolus, triggering a variety of downstream responses that activate p53 leading to cell cycle arrest, senescence and apoptosis [77,78,79] (Figure 5).

Under normal conditions, the tumor suppressor protein p19ARF is expressed at very low levels and is sequestered into the nucleolus due to its association with NPM1 [80]. This prevents ARF’s interaction with Mdm2, which in turn can ubiquitinate p53 thereby promoting its degradation and maintaining it at low levels. In contrast, during replicative senescence or stress induced by activation of oncogenes such as c-MYC and H-RAS, ARF rapidly accumulates, binds to Mdm2 and inhibits its activity, leading to p53 activation (Reviewed in [5,6]). In addition, NPM1 can shuttle to the nucleoplasm and bind directly to Mdm2, independently of ARF, to prevent Mdm2′s function in degrading p53, leading to p53 pathway activation [6,81]. Furthermore, in response to DNA damage, the promyelocytic leukemia (PML) tumor-suppressor protein was shown to sequester Mdm2 in the nucleolus leading to p53 stabilization [82] (Figure 5). When ribosome biogenesis is disrupted at the level of rRNA synthesis, processing or assembly, p53 is induced via an independent pathway, termed the impaired ribosome biogenesis checkpoint (IRBC) [83]. The IRBC is activated upon the release of free ribosomal proteins (RP) L11 (RPL11) and RPL5 to the nucleoplasm in a complex with the 5S rRNA that can sequester and inhibit Mdm2 leading to p53 stabilization (Figure 5). 

Classical chemotherapeutics (e.g., oxaliplatin, cisplatin, actinomycin D and 5-FU) and PARP inhibitors, recently FDA approved for ovarian cancer treatment, have been discovered to act through distinct mechanisms of action that include inhibition of Pol I transcription or ribosome biogenesis [10,70,84]. Indeed, many cancer therapies originally intended to cause DNA damage and kill cancer cells actually impair ribosome biogenesis, which leads to cell cycle arrest and senescence [10,85]. As such the nucleolar stress response serves as a potential therapeutic target for cancer therapy. As it is clear that DDR and ribosome biogenesis do not operate in isolation, more studies into the crosstalk between the DDR, Pol transcription and ribosome biogenesis are required for harnessing the n-DDR as a cancer therapy. In this review, we focus on CX-5461, a novel Pol I transcription inhibitor showing promising activity in clinical investigations [86]. CX-5461 induces n-DDR compared to other Pol I inhibitors such as Actinomycin D and BMH-21, which impair Pol I transcription elongation without inducing DDR [87,88,89]. As this review focusses on CX-5461, we refer readers to reviews on non-selective and selective inhibitors of Pol I transcription [85,90].

## 5. The Nucleolar DDR in Cancer Therapy

In search for next-generation inhibitors that directly target Pol I function, Cylene Pharmaceuticals initially discovered CX-5461 in a molecular-based quantitative real-time polymerase chain reaction (qRT-PCR) screening approach designed to identify and differentiate compounds that selectively inhibit Pol I transcription relative to Pol II transcription [91,92]. Among the numerous molecules screened, CX-5461 was found to inhibit the rate of rDNA transcription in a broad range of cancer cell lines in vitro, exhibiting greater than 200-fold selectivity for Pol I transcription, with limited direct effects on the transcription of Pol II target genes (e.g., c-MYC, ACTB) [91].

CX-5461 has demonstrated single-agent therapeutic efficacy in multiple preclinical cancer models including lymphoma, acute myeloid leukemia (AML), breast, prostate and ovarian cancer [93,94,95,96,97,98,99] and is showing promising activity in early phase clinical trials in blood and solid cancers [86,100]. CX-5461-mediated induction of the p53-dependent IRBC is a major mechanism of response in p53-wild-type tumor cells leading to cell cycle arrest and apoptosis, which seems to be the dominant response to CX-5461 in blood cancer models including lymphoma and AML models. However, in solid cancer models, the cellular response to CX-5461 is independent of p53 status and is associated with cell cycle arrest, senescence and autophagy [91,101].

### 5.1. CX-5461′s Mode of Action

Initial studies have demonstrated that CX-5461′s primary mechanism of action is through impairing the direct binding of the Pol I-specific transcription factor, selectivity factor (SL-1), to the rDNA promoter [91] (Figure 6). SL-1 is a complex containing the TATA-binding protein (TBP) and five TBP-associated factors (TAF1A, TAF1B, TAF1C, TAF1D and TAF12) [102,103]. UBF facilitates the recruitment and binding of SL-1 to the rDNA promoter elements [104], which in turn recruits the Pol I complex [105]. The interaction of transcription factor RRN3 with Pol I is essential for the formation of the transcriptionally competent pre-initiation complex (PIC) [102,106,107]. The DNA topoisomerase TOP2α isoform is an essential component of the initiation-competent Pol I complex where it binds to RRN3 and acts to produce topological changes at the rDNA promoter that in turn promotes PIC formation [108]. Promoter opening and escape are also stimulated by UBF and accompanied by the release of RRN3 from the Pol I complex [109]. CX-5461 inhibits the initiation of Pol I transcription by disrupting the binding of SL-1 to the rDNA promoter [88,91,93,95] and preventing subsequent recruitment of Pol I, leading to exposed rDNA repeats and perturbations in rDNA chromatin structure [88,91,93]. Indeed, recent single-molecule imaging of UBF and Pol I has shown that CX-5461 induces Pol I to dissociate from rDNA chromatin and to move as a liquid within the nucleolar caps [110].

Recent findings have revealed additional modes of action for CX-5461 including blocking of Pol I promoter escape without affecting PIC formation [111], stabilization of nucleolar-associated R-loops and inducing replication stress [97] as well as stabilization of G4 structures [96]. Recently, CX-5461 was also identified to exhibit a sensitivity profile and core chemical structure that resembles known TOP2 poisons, such as etoposide and voreloxin [112]. Type II topoisomerases are enzymes that alter the topological state of DNA by producing temporary DSBs to resolve the positive or negative supercoiling that often arises during replication, transcription and DNA repair, generating a transient cleavage complex through which another DNA helix can pass [113,114,115,116]. Once this strand passage event is complete, the break is re-ligated, and the DNA structure is restored [117,118]. TOP2 poisons such as doxorubicin and etoposide act to stabilize normally transient TOP2-DNA covalent complexes, resulting in the accumulation of cytotoxic DSBs [119,120]. Pipier et al. found that small interfering RNA (siRNA)-mediated depletion of the TOP2α isoform or inhibition of its catalytic activity confers resistance to CX-5461 [121]. Since CX-5461 has been shown to induce activation of ATR at rDNA indicating nucleolar replication stress [97,122], it is plausible that CX-5461 may carry out its action by potentially trapping TOP2α with high selectivity to rDNA promoters compared to regions across the genome. Indeed, a recent study has shown that CX-5461 may act as a DNA structure-driven TOP2-poison at transcribed regions bearing G4 structures [123]. CX-5461 was shown to induce DSBs through a TOP2-dependent mechanism, however, its activity seems to differ from F14512, a selective and potent TOP2A poison. Furthermore, DNA breaks induced by CX-5461 were reduced when Pol I activity was inhibited by another Pol I inhibitor BMH-21 [123], indicating the contribution of rDNA transcription to the cellular response to CX-5461. The data suggest that the interaction of CX-5461 with DNA and sensitivity to CX-5461 are facilitated by DNA topological stress provoked by a high level of Pol I transcription [123,124]. CX-5461-mediated chromatin defects, G4 stabilization or R-loops formation at transcriptionally active loci leads to inhibition of Pol I transcription, mobilizing of TOP2 to resolve topological stresses and subsequent TOP2 poisoning. This model provides a new concept of DNA structure-driven TOP2 poisoning by CX-5461 at rDNA sequences [123]. Altogether, these observations highlight the breadth of CX-5461′s cellular activity mediated through the direct targeting of Pol I transcription and rDNA chromatin.

### 5.2. Therapeutic Response to CX-5461 Is Mediated via Activation of the IRBC and the n-DDR

Using genetically engineered and xenograft models of Eμ-Myc lymphoma, Bywater et al. demonstrated that CX-5461 induces nucleolar disruption, resulting in the binding of unassembled RPL5 and RPL11 to Mdm2 and subsequently rapidly activating p53-mediated apoptosis [93]. Furthermore, this response was triggered in the absence of changes in the total levels of functional ribosomes or the rates of protein synthesis, clearly demonstrating that certain tumor cells are highly dependent on accelerated rDNA transcription levels and intact nucleolar integrity for their survival [83,125]. Furthermore, a strong correlation between the mutational status of p53 and sensitivity to CX-5461 was also found, whereby p53-wildtype lymphoma and AML cell lines were significantly more sensitive towards CX-5461 than p53-null or -mutant cell lines [93,95]. Nevertheless, CX-5461 demonstrated significant efficacy in p53-null AML in vivo. The significant survival advantage in both p53-wildtype and p53-null leukemic mice treated with CX-5461 was associated with an aberrant G2/M cell-cycle progression, induction of myeloid differentiation and targeting of the leukemia-initiating cell population [95].

Insight into the p53-independent effects of CX-5461 was demonstrated in BJ telomerase reverse transcriptase (TERT)-immortalized human fibroblasts expressing a short hairpin RNA targeting p53, where CX-5461 exhibited both p53-dependent and -independent growth inhibitory effects in the absence of global DNA damage [88]. In addition to inhibiting rRNA synthesis, CX-5461 was shown to induce perturbations in rDNA chromatin structure by displacing Pol I from rDNA promoters, thus resulting in the presence of “exposed” rDNA repeats devoid of Pol I (Figure 6). This alteration in rDNA topology was found to trigger non-canonical activation of ATM/ATR kinase signaling within the nucleoli in the absence of detectable γH2AX foci [88], a marker of DSBs, and to promote cell proliferation defects as well as inducing senescence.

CX-5461-mediated inhibition of Pol I transcription in acute lymphoblastic leukemia (ALL) has also been demonstrated to induce a G2/M checkpoint arrest and promote caspase-dependent apoptosis via the ATM/ ATR kinase pathways, independent of p53 status [99,126]. In various models, CX-5461 has been shown to activate ATM/ATR signaling and the combination of CX-5461 with dual inhibition of CHK1/2 has been reported to significantly enhance the therapeutic outcome against p53-null MYC-driven lymphoma in vivo [88].

CX-5461 also activates p53-independent ATM/ATR signaling and the G2/M cell cycle checkpoint in high-grade serous ovarian cancer (HGSOC) cells, leading to growth arrest in vitro and in vivo [97,122]. CX-5461 treatment induces RPA phosphorylation and ATR activation within the nucleoli in HGSOC cells, indicating CX-5461 induces replication stress at the rDNA [97]. The net result of CX-5461-mediated nucleolar DDR is global replication stress. Indeed, CX-5461 treatment is associated with increased levels of S4/S8 phosphorylation of RPA32, a marker of persistent replication stress that is essential for G2M checkpoint activation. Furthermore, CX-5461 has been shown to induce destabilization of replication forks, leading to replication-dependent DNA damage with an overall low induction of γH2AX levels compared with other chemotherapeutics [122]. In HR -deficient cells, CX-5461 enhances nucleolar replication stress, global replication stress and replication-dependent DNA damage, suggesting the HR pathway is required for resolving CX-5461-mediated replication stress [96,97]. Indeed, CX-5461-mediated n-DDR has been shown to be synthetic lethal with HR deficiency [97,122]. Approximately 50% of HGSOC cases are characterized by HR deficiency for which PARP inhibitors have been recently approved as the standard-of-care therapy. CX-5461 cooperates with PARP inhibitors in enhancing replication stress and inhibiting HGSOC cell growth in vitro and in vivo [97] and also in inhibiting tumor growth of HR proficient prostate cancer patient-derived xerographs in vivo [127], highlighting this combination as an exciting approach for ovarian and prostate cancer treatment. Moreover, CX-5461 exhibits single-agent efficacy in patient-derived HGSOC cells harboring a replication fork protection phenotype, a common mechanism of resistance to chemotherapy and PARP inhibitors [97], suggesting that CX-5461 through its p53-independent effects in activating the n-DDR can be harnessed to treat relapsed ovarian cancer.

### 5.3. CX-5461 in Combination with Standard of Care Cancer Therapies

While CX-5461 represents an exciting therapeutic option for multiple cancer types, the identification of predictive biomarkers of response to identify patients who will benefit from this therapy is essential for the success of future clinical trials. In a study utilizing a panel of ovarian cancer cell lines, sensitivity to CX-5461 was shown to be associated with a high baseline rate of Pol I transcription and higher proportion of active to inactive rDNA repeats [124]. This is consistent with CX-5461′s mode of action in inhibiting Pol I transcription and triggering defects associated with open chromatin and replication stress at the rDNA. Therefore, cells with a higher ratio of active rDNA are more sensitive to CX-5461-mediated nucleolar DDR and activation of cell cycle checkpoints. Moreover, biomarkers of sensitivity to CX-5461 in ovarian cancer models include BRCA-mutated and MYC targets gene expression signatures and were found to be enriched in a subset of primary and relapsed ovarian cancer [97]. As MYC is a master regulator of ribosome biogenesis, MYC-driven Pol I transcription and/or MYC-driven global transcription and replication stress may underlie sensitivity to CX-5461.

Importantly, a recent whole protein-coding genome RNAi screen to identify potential targets whose inhibition can enhance the efficacy of CX-5461 in treating HR-proficient HGSOC cells has demonstrated the power of synthetic lethality in the context of CX-5461-induced n-DDR [122]. The primary screen identified 371 genes whose inhibition co-operated with CX-5461 in inhibiting cell growth, of which 41 genes controlled critical DNA repair pathways including HR, replication fork stability, NHEJ, MMEJ and nucleotide excision repair. One of the highly significant CX-5461 synthetic lethal hits was DNA topoisomerase 1 (TOP1), an enzyme that relieves the strain generated by the DNA double helix’s twisting during DNA replication and transcription. TOP1 plays an important role in the rRNA genes [128], and TOP1 inhibitors such as topotecan are used in cancer therapy though their clinical use has been limited due to hematological toxicity. The combination of CX-5461 and low-dose topotecan markedly enhanced n-DDR and global replication stress without increasing DSBs, leading to improved efficacy against HGSOC tumors in vivo [122]. Thus, revealing the potential of harnessing n-DDR in combination with low doses of chemotherapeutics or PARP inhibitors to improve their efficacy in the clinic.

## 6. Perspectives

The field of n-DDR is still in its infancy. Investigating the mechanisms of n-DDR and the crosslink with global DNA repair and DNA replication will bring a fundamental knowledge of how nucleolar surveillance of stress stimuli coordinate activation of cell cycle checkpoints and stress signaling pathways. Characterizing of n-DDR will likely uncover a novel class of non-genotoxic DDR therapeutics with improved efficacy and reduced toxicity compared to DNA damaging chemotherapies.

## Figures and Tables

**Figure 1 genes-12-01156-f001:**
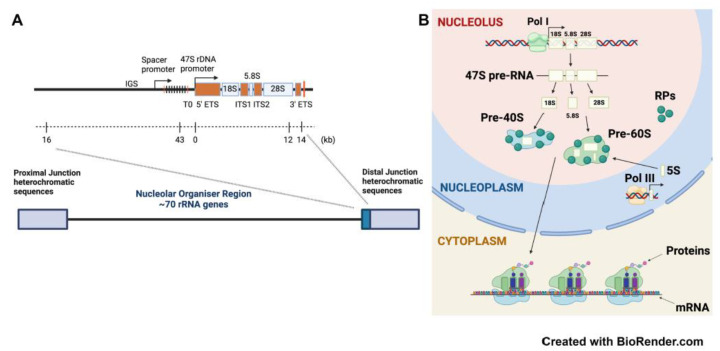
rDNA structure and ribosome biogenesis in mammalian cells. (**A**) Organization of ribosomal RNA (rRNA) genes in mammalian cells. Five NORs, each containing ~70 rRNA genes, are located on the short arms of the acrocentric chromosomes. A canonical rDNA unit consists of the 47S rRNA gene transcribed region made up of the 18S, 5.8S and 28S, coding regions flanked by ETS (external transcribed spacer) and ITS (internal transcribed spacer) elements, and the IGS. The IGS contains regulatory elements including spacer promoter and the 47S rDNA promoter. (**B**) Ribosome biogenesis is a tightly coordinated process involving all three RNA polymerases (Pol I, Pol II and Pol III). RNA Pol I transcribes the tandemly repeated ribosomal RNA (rRNA) genes to produce the 47S precursor rRNA (47S pre-rRNA) transcript in the nucleolus. Following transcription, the 47S pre-rRNA is subsequently cleaved and processed into the mature 18S, 5.8S, and 28S rRNA species. These molecules are then assembled with ribosomal proteins and the 5S rRNA produced by Pol II and III, respectively, to form the small (40S) and the large (60S) ribosomal subunits, which are exported from the nucleolus to the cytoplasm, where they form the mature (80S) ribosome required to initiate mRNA translation and thus protein synthesis.

**Figure 2 genes-12-01156-f002:**
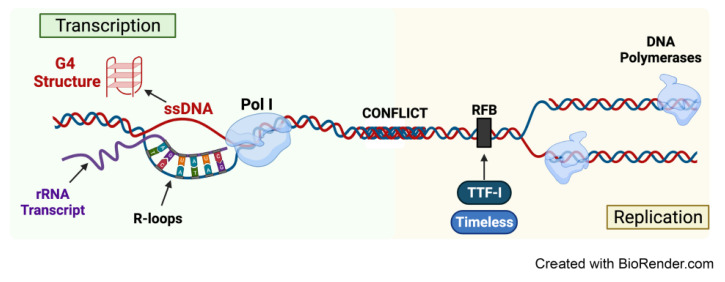
Conflicts between transcription and replication machineries at rDNA. High levels of rDNA transcription cause vulnerability to transcription-replication conflicts. Collisions of Pol I and DNA replication forks trigger positive supercoiling, impinging upon transcription and replication processes at rDNA. Negative supercoiling as a result of unwinding DNA during transcription stabilizes R-loops comprising of stable rRNA transcript and template strand DNA hybrid and displaced single-stranded DNA (ssDNA). G4 structures can form in the displaced ssDNA of R-loops. These unique features drive genomic instability at the rDNA loci. TTF-I and Timeless factor binding to RFB are present as protective mechanisms against potential clashes at early S phase between transcription and replication machineries. Abbreviations: RNA Polymerase I (Pol I), single-stranded DNA (ssDNA), replication fork barriers (RFB), transcription termination factor (TTF-I).

**Figure 3 genes-12-01156-f003:**
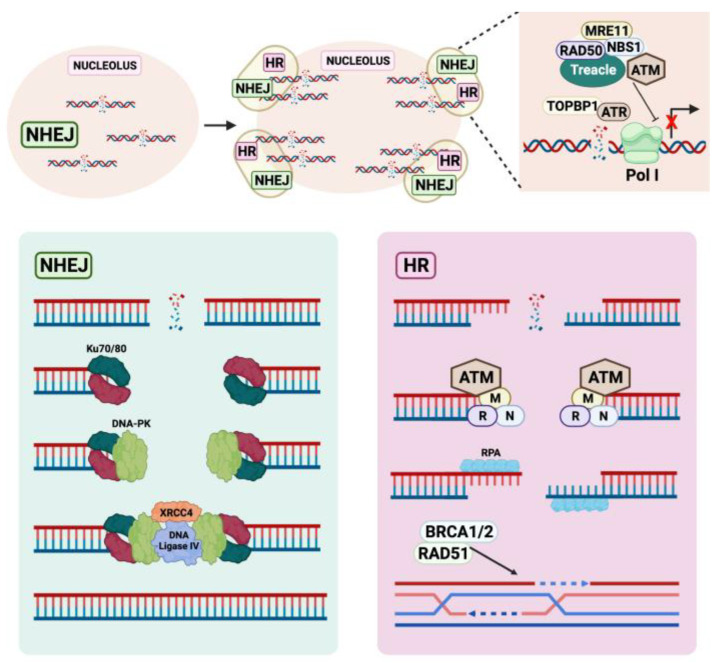
Mechanism of rDNA repair in the nucleolus. Low levels of rDNA damage are typically repaired via the highly mutagenic NHEJ process. It involves direct ligation of broken ends, mediated by several factors including Ku70/80 heterodimer, DNA-PK, DNA ligase IV and XRCC4 complex. Persistent rDNA damage results in ATM-mediated response that involves Treacle, the MRN complex, TOPBP1 and ATR. This leads to Pol I transcription inhibition and translocation to nucleolar caps, where HR repair occurs. HR proteins RPA, RAD51, RAD52, BRCA1 and BRCA2 are present in abundance in the nucleolar caps. Abbreviations: non-homologous end-joining (NHEJ), DNA-dependent protein kinase catalytic subunit (DNA-PK), X-ray repair cross-complementing protein 4 (XRCC4), homologous recombination (HR), ataxia telangiectasia mutated (ATM), ataxia telangiectasia and Rad3-related (ATR), DNA topoisomerase II binding protein 1 (TOPBP1), RNA Polymerase I (Pol I), replication protein A (RPA), breast cancer susceptibility protein 1 and 2 (BRCA1/2).

**Figure 4 genes-12-01156-f004:**
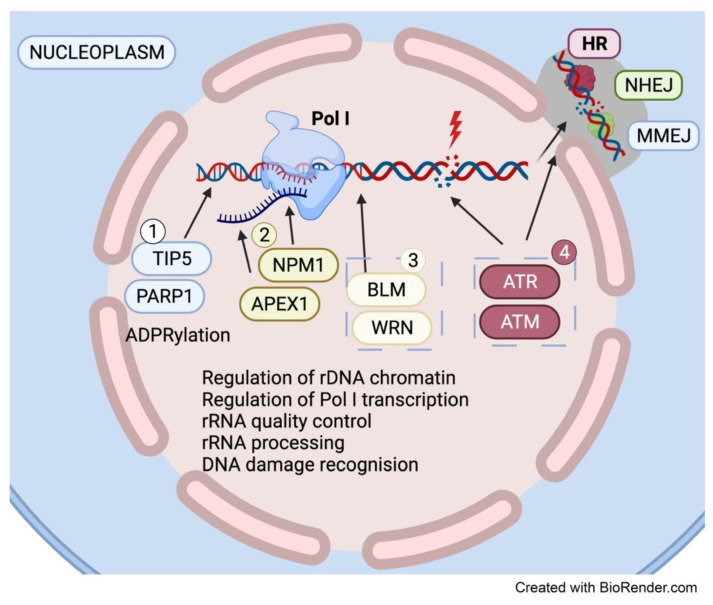
The Nucleolus is a hub for DDR factors. Several DDR factors reside in the nucleolus and may play a dual role in ribosome biogenesis and rDNA repair. (1) PARP1 mediates the PARylation of TIP5 which leads to rDNA silencing and Pol I transcription control. (2) APEX1 interacts with the multi-functional nucleolar protein NPM1 and mediates rDNA repair and rRNA quality control. (3) RecQ-like helicases WRN and BLM reside in the nucleolus and interact with Pol I to facilitate Pol I transcription. (4) Classic DDR factors ATM and ATR are also found in the nucleolus and are involved in an extensive network of factors involved in RNA trafficking, processing, and transcription apart from their role in repairing damaged rDNA.

**Figure 5 genes-12-01156-f005:**
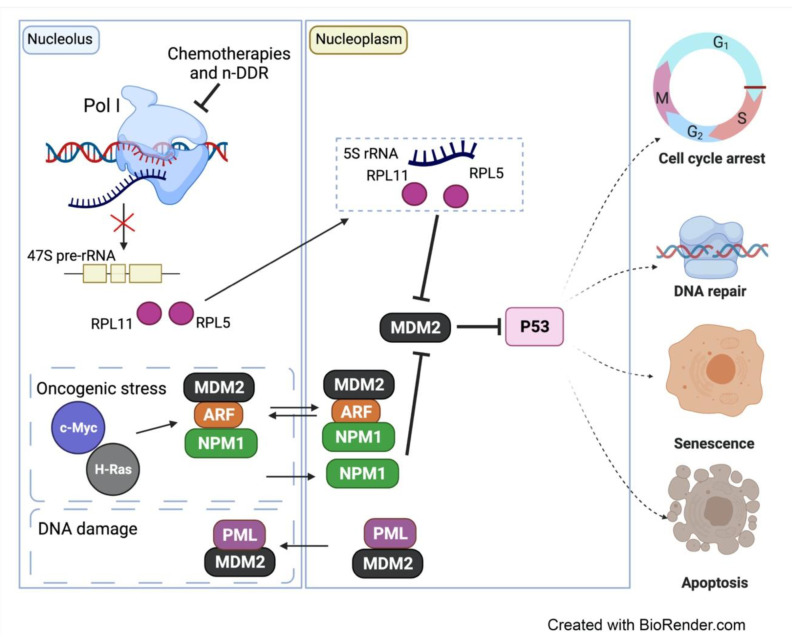
The nucleolar response to different types of stress. Chemotherapies and n-DDR inhibit Pol I transcription and ribosome biogenesis. Free ribosomal proteins, RPL11 and PRL5 then shuttle into nucleoplasm and together with 5S rRNA, they sequester the E3 ligase Mdm2 leading to p53 stabilization. Oncogenic stress also results in the activation of p53 via inhibiting Mdm2. Oncogene such as c-MYC and H-RAS lead to the induction of ARF levels, which together with NPM1 bind and inhibit Mdm2 to activate p53. NPM1 can also interact and inhibit Mdm2 independent of ARF. In response to DNA damage, the PML was also shown to sequester Mdm2 in the nucleolus leading to p53 stabilization. Overall, activation of p53 leads to various responses including cell cycle arrest, DNA repair, senescence, and apoptosis.

**Figure 6 genes-12-01156-f006:**
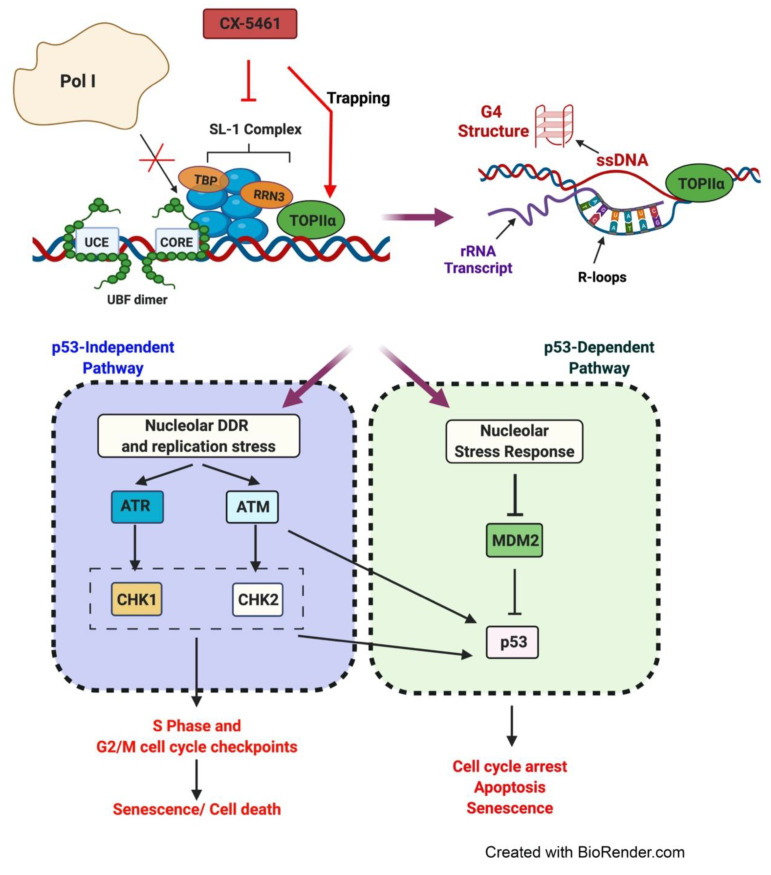
The mechanism of action of CX-5461. CX-5461 inhibits the binding of SL-1 complex to the rDNA promoter region, which in turn prevents the formation of the pre-initiation complex. Furthermore, CX-5461 can induce aberrant rDNA chromatin structures including the formation of single-stranded DNA (ssDNA), stabilization of R-loops and G4 structures, which also leads to inhibition of Pol I transcription, recruitment of TOP2 to resolve topological stresses and subsequent TOP2 poisoning. CX-5461-mediated response involves p53-dependent and -independent pathways. CX-5461 induces activation of the nucleolar stress response which leads to inhibition of Mdm2 and subsequent activation of p53 via multiple pathways. CX-5461 also induces the nucleolar DDR response via activation of ATM and ATR leading to activation of downstream checkpoint kinases CHK1 and CHK2 resulting in cell cycle arrest.

## Data Availability

Not applicable.

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
