# Peer review of "Harnessing the Nucleolar DNA Damage Response in Cancer Therapy"

_genes, 2021, doi:10.3390/genes12081156_

Round 1
Reviewer 1 Report
Xuan et al. prepared a comprehensive review of the nucleolus, nucleolar DDR and cancer treatment using PolI inhibitor. Literature appears to be properly cited and the figures are helpful summary for the readers. This reviewer just have two comments.
- Fig, 5. MDM2 is shown to be sequestered in the nucleolus by ARF (Weber et al. Nature 1999) and by PML (Bernardi, e al., (04)), which is part of the p53 activation mechanisms. In this figure, however, MDM2 is shown in the nucleoplasm. The authors should modify this figure to reflect these studies.
- It is interesting that Bruno et al. claims that pol I inhibition is a secondary effect and is not a primary mode of cytotoxicity by CX-5461. It would be good to discuss this apparent discrepancy by critically analyzing their data dn claim as well as others in the field on this topic, and figure 6 should reflect this possible pol I-independent mechanism.
Author Response
Reviewer#1:
Xuan et al. prepared a comprehensive review of the nucleolus, nucleolar DDR and cancer treatment using PolI inhibitor. Literature appears to be properly cited and the figures are helpful summary for the readers.
We thank the reviewer for their support and encouraging comments.
This reviewer just have two comments.
- Fig, 5. MDM2 is shown to be sequestered in the nucleolus by ARF (Weber et al. Nature 1999) and by PML (Bernardi, e al., (04)), which is part of the p53 activation mechanisms. In this figure, however, MDM2 is shown in the nucleoplasm. The authors should modify this figure to reflect these studies.
We have updated Figure 5 to include binding of Mdm2 and ARF in the nucleolus and the nucleoplasm. As the reviewer points out Weber et al. have shown that ARF sequesters Mdm2 in the nucleolus. However, it is not clear if relocalization of Mdm2 to the nucleolus is necessary to prevent its interactions with p53, or if it is sufficient for ARF to inhibit the ubiquitin ligase activity of Mdm2 in the nucleoplasm (Kruse and Gu, 2009, Lee and Gu, 2010). Furthermore, the C-terminal region of ARF, which includes the nucleolar localization sequence, was shown to be not essential for the regulation of p53 ubiquitinylation by Mdm2 (Zhang and Xiong, 1999).
We have therefore updated the figure and edited the text in line 361 to the following sentence “In contrast, during replicative senescence or stress induced by activation of oncogenes such as c-MYC and H-RAS, ARF rapidly accumulates, binds to Mdm2 and inhibits its activity, leading to p53 activation (Reviewed in [5,6]).”
We have also added a description of PML’s role in activating p53 in line 366 and updated the figure accordingly. “Furthermore, in response to DNA damage, the promyelocytic leukemia (PML) tumor-suppressor protein was shown to sequester Mdm2 in the nucleolus leading to p53 stabilization [82](Figure 5)”.
- It is interesting that Bruno et al. claims that pol I inhibition is a secondary effect and is not a primary mode of cytotoxicity by CX-5461. It would be good to discuss this apparent discrepancy by critically analyzing their data dn claim as well as others in the field on this topic, and figure 6 should reflect this possible pol I-independent mechanism.
We thank the reviewer for their insightful comment. Indeed, studies by Bruno et al., and more recently Bossaert et al 2021 have shown that that CX-5461 may act as act as a DNA structure-driven TOP2-poison at transcribed regions bearing G4 structures. Furthermore, DNA breaks induced by CX-5461 were strongly reduced when Pol I activity was inhibited by another Pol I inhibitor BMH-21, indicating the contribution of rDNA transcription to the cellular response to CX-5461. The data suggest that the interaction of CX-5461 with DNA is facilitated by DNA topological stress provoked by high level of Pol I transcription, that recruits and facilitates TOP2 poisoning. We have updated the review and Figure 6 to include the study described recently by Bossaert et al.
Line 452: “Indeed, a recent study have shown that CX-5461 may act as a DNA structure-driven TOP2-poison at transcribed regions bearing G4 structures [123]. CX-5461 was shown to induce DSBs through a TOP2-dependent mechanism, however its activity seems to differ to F14512, a selective and potent TOP2A poison. Furthermore, DNA breaks induced by CX-5461 were reduced when Pol I activity was inhibited by another Pol I inhibitor BMH-21 [123], indicating the contribution of rDNA transcription to the cellular response to CX-5461. The data suggest that the interaction of CX-5461 with DNA and sensitivity to CX-5461 is facilitated by DNA topological stress provoked by high level of Pol I transcription [123, 124]. CX-5461-mediated chromatin defects, G4 stabilization or R-loops formation at transcriptionally active loci leads to inhibition of Pol I transcription, mobilizing of TOP2 to resolve topological stresses and subsequent TOP2 poisoning. This model provides a new concept of DNA structure-driven TOP2 poisoning by CX-5461 at rDNA sequences [123].”

Reviewer 2 Report
The nucleolus plays an important role in ribosome biogenesis. Given that it undergoes high transcription and contains extensive DNA repeats it is a hotspot for DNA damage. It also appears to have a central role in mediating the p53-driven response to cellular stresses. For these reasons, nucleolar functions are potential anti-cancer therapeutic targets. In this manuscript, the authors present a review of nucleolar biology with respect to genome instability and cancer and detail the recent efforts to disrupt nucleolar function to affect cancer cell killing with the small molecule CX-5461.
The authors present a very detailed background on rDNA structure and function. They go on to explain how genomic instability can occur at rDNA loci through replication/transcription clashes, as well as R-loop and G4 DNA formation. The authors then describe the complex relationship of rDNA structure and function to tumorigenesis in which some cancer exhibit high rDNA CNVs while others have low CNVs. They describe the interplay of the double strand repairing pathways NHEJ and HR at DNA damage within the rDNA and describe the data that demonstrates that the nucleolus is a nexus for DNA repair factors. Finally, the authors then review the anti-polI inhibitor CX-5461. CX-5461 disrupts the interaction of SL-1 with the rDNA promoter region. They summarize how CX-5461 appears to have a myriad of effects on cancer cells that are both p53-dependent and independent. Lastly they describe how CX-5461 could be used in combination with standard of care therapeutics such as topoisomerase inhibitors.
Overall the concepts presented in the review are well laid out, the referencing of primary data appears to be thorough and the manuscript is well-written. Although not necessary, it would have been informative to include mention of related compounds, such as CX-3543 or BMH-21, and and their relationship to nucleolar function and targeting. Similarly, no other options to target the genome instability of nucleolus are presented. Are there other approaches? As such the review is very much about CX-5461 perhaps the title should be amended to reflect that focus.
Minor points:
line 164 “... rDNA copy numbers is several types of cancers …” numbers in several”
line 325 “treacle” - “Treacle”
DOI are missing for ref 7, 88, 89 and 129 (although it is listed as accepted so perhaps doesn't have one).
The author list on 129 also appears odd with "Investigators" at the end of line 838.
Ref 14 appears to be garbled.
Ref 4, 5 and 11 have no assigned sources(journals/websites).
There may be other issues in the references that I have missed.
Author Response
Reviewer 2:
Overall the concepts presented in the review are well laid out, the referencing of primary data appears to be thorough and the manuscript is well-written. Although not necessary, it would have been informative to include mention of related compounds, such as CX-3543 or BMH-21, and their relationship to nucleolar function and targeting. Similarly, no other options to target the genome instability of nucleolus are presented. Are there other approaches? As such the review is very much about CX-5461 perhaps the title should be amended to reflect that focus.
We thank the reviewer for their support and constructive comments. We have focused on CX-5461 due to its ability induce the nucleolar DNA damage response and being the most advanced in the clinic. To clarify our focus on CX-5461, we have added to the abstract, “In this review, we focus on CX-5461, a novel inhibitor of Pol I transcription that induces the nucleolar DNA damage response and is showing increasing promise in clinical investigations.”
In section 4, we also refer to other reviews on non-selective and selective inhibitors of Pol I transcription, line 383: “In this review, we focus on CX-5461, a novel Pol I transcription inhibitor showing promising activity in clinical investigations [86]. CX-5461 induces n-DDR compared to other Pol I inhibitors such as Actinomycin D and BMH-21, which impair Pol I transcription elongation without inducing DDR [87-89]. As this review focusses on CX-5461, we refer readers to reviews on non-selective and selective inhibitors of Pol I transcription [85,90].”
Minor points:
line 164 “... rDNA copy numbers is several types of cancers …” numbers in several”
line 325 “treacle” - “Treacle”
DOI are missing for ref 7, 88, 89 and 129 (although it is listed as accepted so perhaps doesn't have one).
.....There may be other issues in the references that I have missed.
We sincerely thank the reviewer for their comments. We have edits minor typos and corrected the reference style.
